# Development and validation of a minimum requirements checklist for snakebite envenoming treatment in the Brazilian Amazonia

Thiago Serrão-Pinto[1,2,3], Eleanor Strand[4], Gisele Rocha[1,2], André Sachett[1,2], Joseir Saturnino[1,2], Altair Seabra de Farias[1,2], Aline Alencar[1,2], José Diego Brito-Sousa[1,2], Anna Tupetz[4], Flávia Ramos[1,2,5], Elizabeth Teixeira[1], Catherine Staton[4], João Vissoci[4], Charles J. Gerardo[4], Fan Hui Wen[6], Jacqueline Sachett[1,2,7], Wuelton M. Monteiro[1,2]*

1 Escola Superior de Ciências da Saúde, Universidade do Estado do Amazonas, Manaus, Brazil, 2 Diretoria de Ensino e Pesquisa, Fundação de Medicina Tropical Dr. Heitor Vieira Dourado, Manaus, Brazil, 3 Faculdade de Ciências Farmacêuticas, Universidade Federal do Amazonas, Manaus, Brazil, 4 Department of Emergency Medicine, Duke University School of Medicine, Durham, North Carolina, United States of America, 5 Department of Nursing, Universidade Federal de Santa Catarina, Florianópolis, Brazil, 6 Instituto Butantan, São Paulo, São Paulo, Brazil, 7 Diretoria de Ensino e Pesquisa, Fundação Alfredo da Matta, Manaus, Brazil

* wueltonmm@gmail.com

**Data Availability Statement:** All data underlying our findings are fully available in the manuscript files.

## Abstract

### Background

Currently, antivenoms are the only specific treatment available for snakebite envenoming. In Brazil, over 30% of patients cannot access antivenom within its critical care window. Researchers have therefore proposed decentralizing to community health centers to decrease time-to-care and improve morbidity and mortality. Currently, there is no evidence-based method to evaluate the capacity of health units for antivenom treatment, nor what the absolute minimum supplies and staff are necessary for safe and effective antivenom administration and clinical management.

### Methods

This study utilized a modified-Delphi approach to develop and validate a checklist to evaluate the minimum requirements for health units to adequately treat snakebite envenoming in the Amazon region of Brazil. The modified-Delphi approach consisted of four rounds: 1) iterative development of preliminary checklist by expert steering committee; 2) controlled feedback on preliminary checklist via expert judge survey; 3) two-phase nominal group technique with new expert judges to resolve pending items; and 4) checklist finalization and closing criteria by expert steering committee. The measure of agreement selected for this study was percent agreement defined a priori as ≥75%.

**Funding:** J.S. and W.M. were funded by Conselho Nacional de Desenvolvimento Científico e Tecnológico (CNPq productivity scholarships). W. M. was funded by Fundação de Amparo à Pesquisa do Estado do Amazonas (PRÓ-ESTADO, call 011/ 2021 - PCGP/FAPEAM, call 010/2021 - CT&I ÁREAS PRIORITÁRIAS, call 003/2022 - FAPEAM, and POSGRAD/FAPEAM, and POSGRAD RESOLUÇÃO N. 002/2023) and by the Ministry of Health, Brazil (proposal No. 733781/19-035). Research reported in this publication was partly supported by the Fogarty International Center of the National Institutes of Health under Award Number R21TW011944. The content is solely the responsibility of the authors and does not necessarily represent the official views of the National Institutes of Health. The funders had no role in study design, data collection and analysis, decision to publish, or preparation of the manuscript.

**Competing interests:** The authors have declared that no competing interests exist.

## Results

A valid, reliable, and feasible checklist was developed. The development process highlighted three key findings: (1) the definition of community health centers and its list of essential items by expert judges is consistent with the Brazilian Ministry of Health, WHO snakebite strategic plan, and a general snakebite capacity guideline in India (internal validity), (2) the list of essential items for antivenom administration and clinical management is feasible and aligns with the literature regarding clinical care (reliability), and (3) engagement of local experts is critical to developing and implementing an antivenom decentralization strategy (feasibility).

## Conclusion

This study joins an international set of evidence advocating for decentralization, adding value in its definition of essential care items; identification of training needs across the care continuum; and demonstration of the validity, reliability, and feasibility provided by engaging local experts. Specific to Brazil, further added value comes in the potential use of the checklist for health unit accreditation as well as its applications to logistics and resource distribution. Future research priorities should apply this checklist to health units in the Amazon region of Brazil to determine which community health centers are or could be capable of receiving antivenom and translate this expert-driven checklist and approach to snakebite care in other settings or other diseases in low-resource settings.

### Author summary

Checklists have been developed and validated to improve patient safety and effectiveness of care in several fields, including emergency medicine, intensive care, and surgery. The Brazilian Ministry of Health (MoH) supplies antivenoms (AVs) to the health system at no cost to patients. AV access is thus limited to hospitals, most of which are in urban areas and difficult for rural, remote, and indigenous populations to reach. Currently, there is no evidence-based method to evaluate the capacity of health units for AV treatment, nor what the absolute minimum supplies and staff are necessary for safe and effective AV administration and clinical management. In this study, we aim to develop and validate a checklist to evaluate the minimum requirements for community health centers to adequately treat snakebite envenoming in the Amazon region of Brazil. This study joins an international set of evidence advocating for decentralization, adding value in its definition of essential care items, represented by Human Resources, and Equipment, Supplies and Medicines, to provide safe and effective treatment for SBE patients in remote endemic areas.

## Introduction

In the Brazilian Amazon, the incidence of snakebite envenoming (SBE), though underestimated, is roughly 30,000 cases per annum [1]. Currently, antivenoms are the only specific treatment available for SBE. The clinical care window for antivenom (AV) effectiveness is six hours [1]. Antivenom administered after this window is less effective in reversing systemic

damage and has little or no effect on local tissue damage [2]. Timely access to AV is thus crucial to avoid preventable complications, disabilities, and death [3]. In Amazonia, Brazil, however, over 30% of snakebite patients cannot access AV within six hours [4].

This is due in part to the AV distribution structure. Currently, the Brazilian Ministry of Health (MoH) supplies AVs to the health system at no cost to patients [5]. The MoH first distributes AVs to each state. State health secretaries are responsible for distributing AVs to municipalities, and municipalities then distribute AVs to their hospitals. AV access is thus limited to hospitals, most of which are in urban areas and difficult for rural, remote, and indigenous populations to reach. The national health care system in Brazil, Sistema Único de Saúde (SUS), provides access to care to these populations through community health centers (CHC).

CHCs provide primary care to designated populations [6]. In the remote and rural regions of Amazonia, where snakebites tend to occur, CHCs are often the only health facility [4]. To reduce the delay in antivenom treatment and improve patient outcomes, clinical and research experts in the State of Amazonas have called for a decentralization strategy in which antivenom is supplied to CHCs in addition to hospitals [4,7,8].

Our research team has been utilizing data and implementation science to establish a comprehensive understanding of SBE in the Brazilian Amazonia and develop such a strategy for decentralizing AV treatment to CHCs [1,4,5,7,9–17] (Fig 1). We have proposed: 1) developing and validating a culturally relevant clinical practice guideline, 2) training health professionals according to the guideline, 3) implementing the standardized protocol, 4) assessing perceptions and acceptability of the changes, and 5) estimating the impact on timely AV access. These steps, however, rest on the assumption that CHCs are well-equipped and staffed with personnel trained in administering AV treatment. Currently, there is no evidence-based method to evaluate the capacity of health units for AV treatment [18], nor what the absolute minimum supplies and staff are necessary for safe and effective AV administration and clinical management [8,19].

Checklists, specifically criteria of merit checklists [20], have been developed and validated to improve patient safety and effectiveness of care in several fields [21,22], including emergency medicine, intensive care, and surgical disciplines [23–26]. In Brazil, health authorities have established several checklists to ensure adequate capacity and quality of health services: hospitals [27], community health centers [28], surgical services [29], vaccination services [30], intensive care units [31], clinical laboratories [32], and pharmacies [33]. In terms of antivenom decentralization to CHCs, a criteria of merit checklist for the staff and supplies involved in antivenom treatment and subsequent clinical management is needed to determine which CHCs are or could be capable of receiving antivenom, and to guide authorities in accrediting health units in Brazil to perform these procedures.

Our objective was thus to develop and validate a checklist to evaluate the minimum requirements for community health centers to adequately treat snakebite envenoming in the Amazon region of Brazil.

## Methods

### Ethics statement

The study was approved by the Research Ethics Committee, Fundação de Medicina Tropical Dr. Heitor Vieira Dourado (FMT-HVD; CAAE: 52735721.7.0000.0005, approved on 5 November 2021). Written informed consent was obtained from all participants of the study.

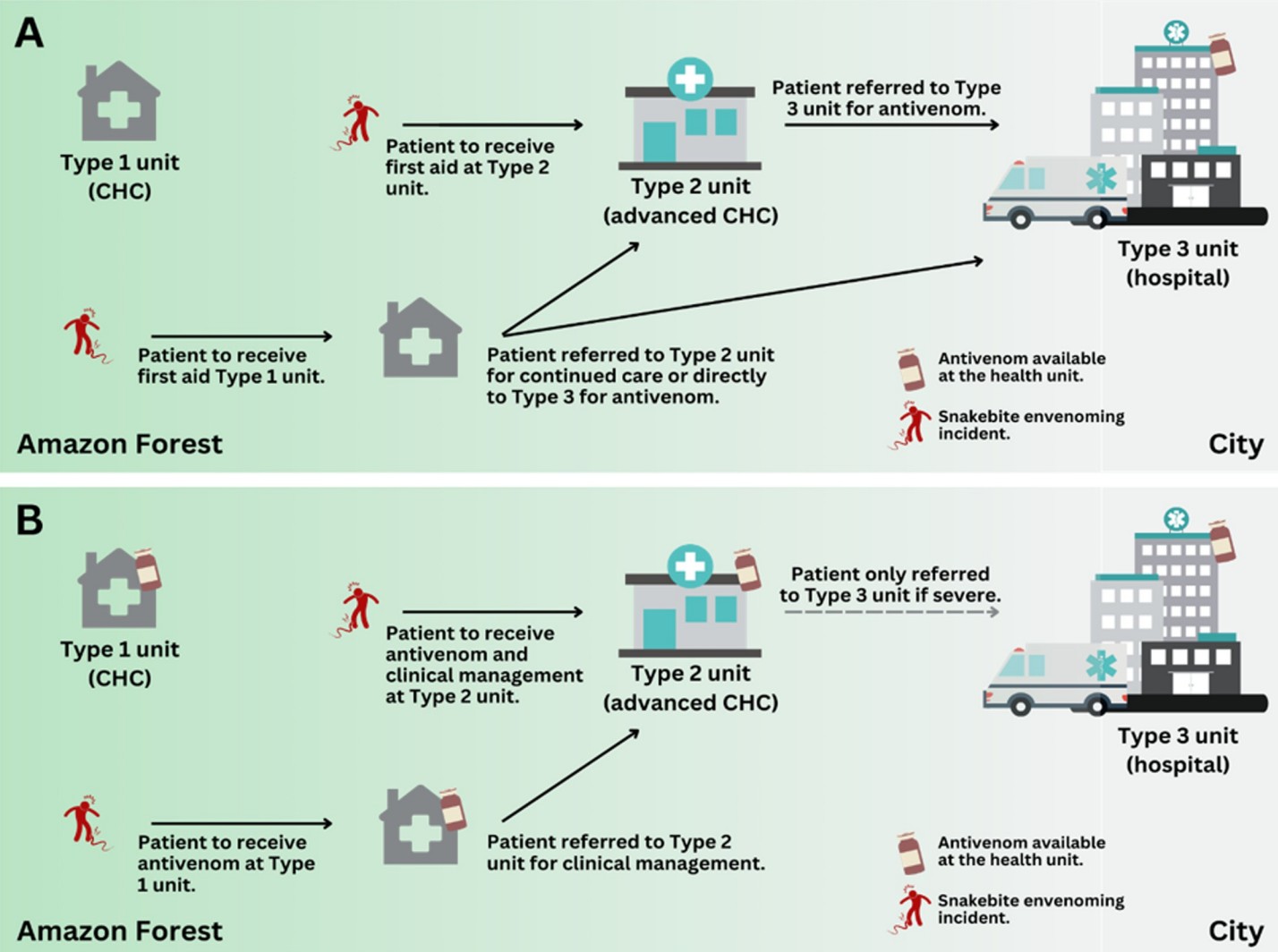

**Fig 1. Snakebite envenoming care continuum in the Brazilian Amazon.** A) Care pathways for snakebite patients under the current antivenom distribution structure. B) Care pathways under the proposed antivenom decentralization to community health centers (CHCs). Figure built using https://openclipart.org/ as source of the images or icons.

## Study design

A modified-Delphi approach was utilized as we contextualized the study to a specific neglected health issue in a specific setting, and the results were time-sensitive in their direct application to the development and implementation of an antivenom decentralization strategy [34]. The checklist was developed in accordance with the best practices outlined by Bichelmeyer, Scriven, and Stufflebeam [20,35,36] to define the minimum requirements for community health centers to adequately treat snakebite envenoming in the Amazon region of Brazil. The modified-Delphi approach consisted of four rounds: 1) iterative development of preliminary checklist by expert steering committee; 2) controlled feedback on preliminary checklist via expert judge survey; 3) two-phase nominal group technique [37,38] with new expert judges to resolve pending items; and 4) checklist finalization and closing criteria by expert steering committee (Fig 2).

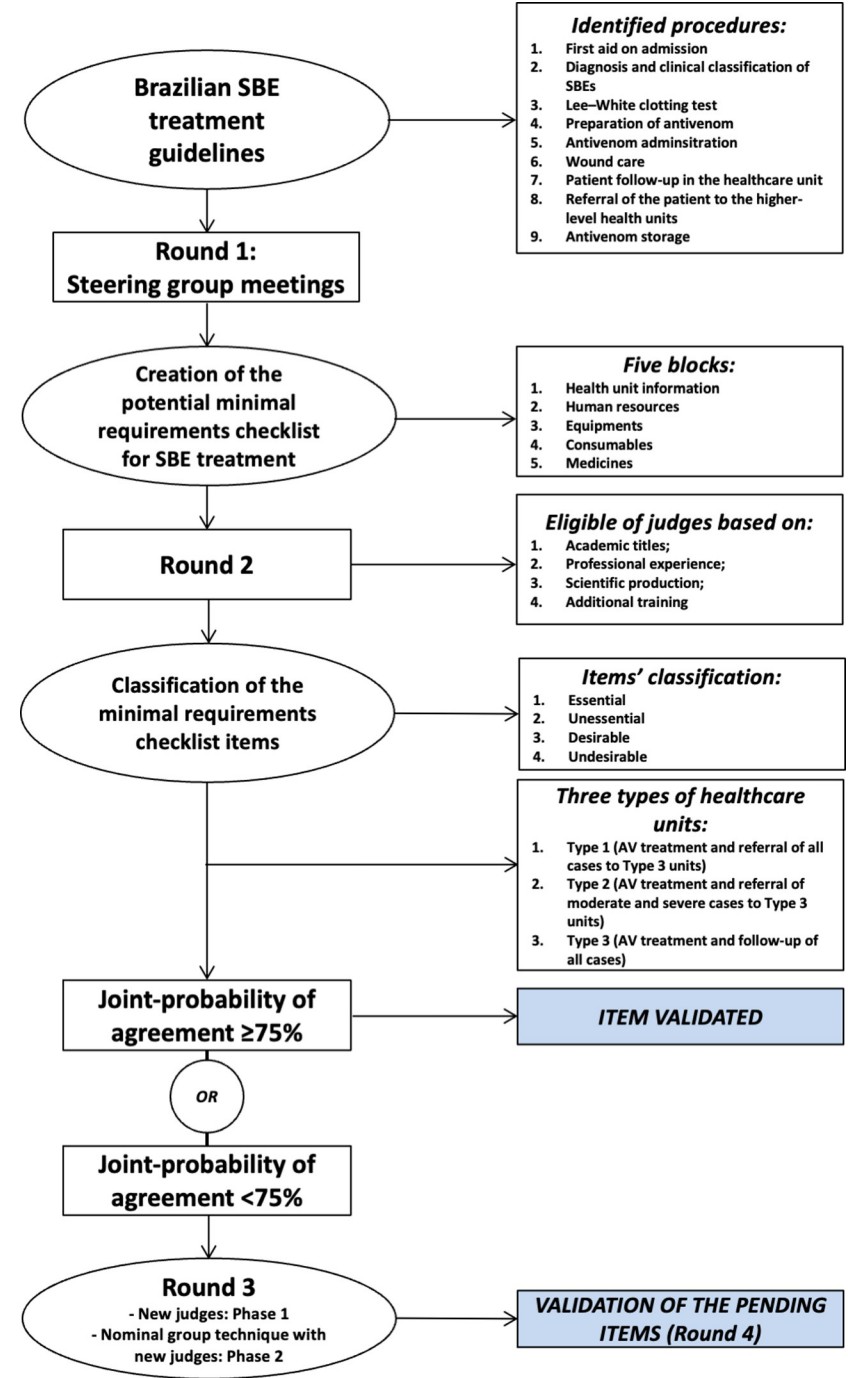

**Fig 2. Study design overview.**

## Measure of agreement

The measure of agreement selected for this study was percent agreement, specifically joint-probability of agreement (JPA) [39]. For each type of community health center (1, 2, or 3), JPA was calculated by the sum of agreement on an item marked into a specific priority classification (Essential, Unessential, Desirable, or Undesirable), divided by the total responses,

multiplied by 100. Consensus on an item was defined a priori as a JPA greater than or equal to 75%. Items with a JPA ≥75% in Essential or Desirable were considered validated. Items marked into Unessential or Undesirable with a JPA of ≥75% were dropped. Items with a JPA <75% in all four classifications were discussed in a nominal group meeting and either validated or dropped. Additional items suggested by judges in the survey round were discussed in the nominal group meeting, classified by priority, and either added or dropped. The nominal group meeting, and thus the modified-Delphi process, ended when consensus was reached on all items.

Across the Delphi process, three different types of community health centers as well as item priority (i.e., unessential versus essential to adequate treatment) were considered. The primary outcome was measure of agreement, specifically percent agreement.

### Expert selection and recruitment

**Expert steering committee.** A steering committee of five experts were recruited via email to develop the initial checklist, coordinate consensus rounds, and verify the finalized version. Two experts with postdoctoral degrees and extensive experience in snakebite clinical care and research, a male pharmacist (W.M.) and a female nurse (J.S.), led this effort from the FMT-HVD, a tertiary care hospital providing SBE care to patients across the Brazilian Amazon. One expert with a medical and postdoctoral degree (F.H.W), also highly experienced with SBE clinical care and research, was recruited from the Butantan Institute, the largest antivenom producer and distributor in the country. Two experts with medical degrees and global experience treating and researching snakebite envenoming (C.S., C.G.) were recruited from the Department of Emergency Medicine of the Duke School of Medicine in the United States. These team members have previously collaborated to develop a culturally relevant clinical practice guideline for snakebite treatment in Brazil as well as a multimodal health system intervention to decentralize antivenom from hospitals to community health centers in the Amazon region.

**Expert judges.** To obtain a culturally and geographically diverse panel of judges, potential experts were identified via four lines of inquiry: 1) professors of graduate programs in Tropical Medicine and Clinical Toxicology, provided their lines of research were related to snakebite envenoming in Brazil; 2) a MEDLINE search on snakebite envenoming and related topics over the past five years; 3) health professionals and clinical care directors working at tertiary hospitals; and 4) public health coordinators from the Ministry of Health in Brazil. These potential experts were invited to participate via email and asked to recommend additional experts. Potential experts recommended through this snowball method were also invited to participate. Email invitations to all potential experts included a study description, an informed consent form, and a survey composed of a demographic questionnaire and the first version of the checklist. If there was no response within 20 days of the email invitation or the survey was not fully completed within 20 days of signing the consent form, the potential expert was not included as an expert judge in the survey round.

Potential experts from the survey round who did not respond, refused survey participation, or did not complete the survey fully were emailed a second time and invited to participate in the nominal group meeting. New potential experts identified via snowball sampling were also invited via email to the nominal group meeting. The email invitation for the nominal group meeting included a study description, informed consent form, and a demographic questionnaire. If there was no response within 20 days of the invitation, the potential expert was not included in the nominal group meeting. All potential experts were scored based on the criteria outlined in Table 1. Potential experts were excluded if their score was below five points.

**Table 1. Eligibility criteria for expert judges (≥5 points).**

| Criteria | Description | Points |
|---|---|---|
| Academic training | Doctorate in subject area | 3 |
| | Master in subject area | 2 |
| | Specialist / medical residency in subject area | 1 |
| Additional training | Participation in training courses in the subject area | 1 |
| Professional experience | Minimum of two years patient care in the subject area | 2 |
| | Minimum of two years teaching in the subject area | 2 |
| Scientific production | Dissertation, thesis. or monography in the subject area | 1 |
| | Papers published related to the specific area | 2 |
| | Supervision of students in the subject area | 1 |

## Research team and reflexivity

The research team included the expert steering committee (W.M., J.S., F.H.W, C.S., C.G.); four validation specialists (E.S., G.S., E.T., F.R.); a qualitative specialists (J.S.) with nursing experience in clinical SBE care and research; a specialist in indigenous health (A.S.F.), a clinical laboratory scientist (J.D.B.-S.); a qualitative researcher (E.S.); a licensed physical therapist and qualitative data specialist with experience in SBE research (A.T.); a psychologist with extensive qualitative and quantitative data analytics training and experience (J.V.); a clinical pharmacist (T.S.P.); a clinical researcher (A.A.); and a data management specialist (A.S.).

## Modified-Delphi method

**Round 1: Iterative development of preliminary checklist by expert steering committee.** The steering committee developed the first version of the checklist in three, iterative online meetings spaced one week apart. In the first meeting, items were listed in a Word file in a brainstorming process along with a basic description. This process was structured according to the SBE clinical management and antivenom administration procedures in the clinical practice guideline from the Ministry of Health [40] as well as the recently developed clinical practice guideline specifically for community health centers [10]. Items were thus listed for the categories outlined in these guidelines: first aid on admission; diagnosis (*Bothrops*, *Lachesis*, *Crotalus*, and *Micrurus*) and clinical classification (mild, moderate, or severe) of SBEs; Lee–White clotting test procedure; preparation of antivenom before administration; antivenom administration; wound care; patient follow-up during the stay in the healthcare unit; referral of the patient to the higher-level health units, if necessary; and receiving and storing antivenoms. At the end of the first meeting, the list of items was shared with committee members to review over the week. In the second meeting, the committee discussed the list. Any additional items thought of during the week were suggested and included in the checklist. The committee was provided another week to review the list. In the third meeting, the committee finalized the preliminary version of the checklist and added more detailed descriptions of each item and its role in snakebite envenoming care (S1 File).

**Round 2: Controlled feedback on preliminary checklist via expert judge survey.** Expert judges received a survey to provide controlled feedback on the preliminary checklist. The survey was divided into three sections: 1) judge demographics, 2) context for the checklist, and 3) the preliminary version of the checklist. Demographics included gender, age, profession, experience, and education. The context section situated expert judges within the three types of health units (Table 2) and four levels of priority (Table 3).

**Table 2. Types of health units according to capacity for snakebite management.**

| Unit capacity | Description |
|---|---|
| Type 1 (CHCs) | Type 1 units are community health centers able to provide basic first aid and administer antivenom but refer all patients to Type 2 units for clinical management. In severe cases, Type 1 units could refer patients directly to Type 3 (hospitals). |
| Type 2 (advanced CHCs) | Type 2 units are community health centers that function as an intermediate level of care with additional equipment/supplies, larger infrastructure, and specific training in emergency response compared to Type 1. Type 2 units should have the capacity to administer antivenom (if not previously given by Type 1 providers; for example, the Type 2 unit is the first point of contact), as well as provide clinical management, observation of potential adverse reactions, and follow-up care to patients. Type 2 units would refer severe cases to Type 3 (hospitals). |
| Type 3 (hospital) | Type 3 units are hospitals capable of treating all snakebite envenoming cases, including severe cases. |

For each type of health unit, expert judges classified items by priority. Expert judges were also provided with a free text field to suggest additional items and/or provide comments to improve the checklist. Items with a JPA ≥75% in Essential or Desirable were considered validated. Items marked into Unessential or Undesirable with a JPA of ≥75% were dropped. Items with a JPA <75% in all four classifications as well as items suggested by expert judges in the free text portion of the survey were discussed in the next round.

**Round 3: Two-phase nominal group technique with new expert judges.** New expert judges, independent from Round 2, attended a two-phase nominal group meeting [37,38]. The aim of the meeting was to reach consensus on items with a JPA <75% and items suggested by the expert judges in the previous round. A lead member of the expert steering group (W.M.) and a graduate student observer (T.S.P) conducted this meeting. In the first phase of the meeting, expert judges classified items by the four priority levels for each type of health unit. Responses were collected anonymously by the facilitators (W.M, T.S.P) and organized on a spreadsheet. JPA was calculated for each item, then fed back to expert judges. The second phase of the meeting was discussion until consensus regarding the items with a JPA <75%. Up to two additional nominal group meetings were planned to reach consensus and close Round 3.

**Round 4: Checklist finalization and closing criteria by expert steering committee.** The expert steering committee produced a final version of the minimum requirements checklist based on the previous three rounds of development and controlled feedback. Items considered Unessential with a JPA ≥75% were not included in the checklist. Items considered Undesirable with a JPA ≥75% were also not included in the checklist, but noted at the end in a warnings section. The checklist was finalized according to best practices for checklist development, validation, and practical use outlined by Bichelmeyer, Scriven, and Stufflebeam [20,35,36].

**Table 3. Priority classifications of items.**

| Priority | Description |
|---|---|
| Essential | Presence of the item is mandatory for antivenom storage and administration to the patient |
| Desirable | Presence of the item is not mandatory, but improves the quality of antivenom storage and administration to the patient, offering greater convenience and comfort for health professionals and patients, respectively |
| Unessential | Presence of the item is indifferent for antivenom storage and administration to the patient |
| Undesirable | Presence of the item can be harmful for antivenom storage and administration to the patient |

## Results

Results are presented according to each modified-Delphi round. A summary of expert judge responses from Round 2 and Round 3 is provided in Fig 3.

### Round 1: Iterative development of preliminary checklist

The expert steering committee structured the first version of the checklist in five sections (Table 4). The first section, Health Unit Information, includes items regarding identification information of the community health center as well as its basic infrastructure and capacity. The second section, Human Resources, outlines personnel capacity and availability across different professional categories: nursing technicians and nurse assistants, nurses, physicians, laboratory personnel, and pharmacists. The remaining three sections list the Equipment, Supplies, and Medicines utilized in snakebite envenoming care.

Prior to completing the first version of the checklist, the expert steering committee discussed the priority of some items. The presence of electricity and the ability to transfer patients were defined as Essential for all three types of health units. Electricity is required to store antivenom, as it is refrigerated, and the capacity for patient transfer to hospitals is necessary in severe cases. Items required for the 20-minute whole blood clotting test (necessary for determining antivenom indication) were defined as Essential for all units. The ability of a health unit to operate 24/7 was considered desirable in Type 1 community health centers (most basic care), and essential in Type 2 (more advanced CHC) and Type 3 (hospitals) units.

### Round 2: Expert judge survey of preliminary checklist

**Participant characteristics.**  A total of 35 potential experts were invited to participate in the survey round. Nine invitations were not returned and two were refused. A total of 24 invitations were thus accepted. Two of these potential experts did not fully complete the survey and were excluded. An additional two were excluded based on the minimum criteria for participation as experts.

A total of 20 potential experts were included as expert judges (Table 5). The majority were nurses (75%), men (60%), and from Amazonas State (50%). The average age was 44.75 years. Professional experience averaged almost 12 years, with most expert judges holding advanced degrees (70%), previous or current professor positions in universities (85%), published articles (75%), and clinical SBE care experience (95%). The average expert criteria score was 10.90. See S2 File for additional characteristics.

**Survey of preliminary checklist.**  All items in the Health Unit Information section were validated for all types of health units. No item suggestions were made by the expert judges. For Type 1 community health centers, less than half the items in the remaining sections (42.0%) were validated. Most items, however, were validated for Type 2 (76.8%) and Type 3 (91.3%). All items validated in this round were classified as Essential. See S3 File for each item classifications and its JPA for the three types of health units. Overall, the items in Supplies (39.3%) and Medicines (41.7%) sections had the lowest JPA in Type 1. The Equipment section had the lowest JPA in Type 2 (75.0%) and Type 3 (87.5%) centers.

In terms of free text recommendations, the expert judges suggested the addition of 11 items. A reclining stretcher, multiparameter patient monitor, and hospital screen were added in the Equipment section. A central venous catheter, indwelling urinary catheter, laryngeal mask airway, and urine collection bag were added in Supplies. Anticonvulsants, antiemetics, atropine, and bicarbonate were added in Medicines. No items were dropped.

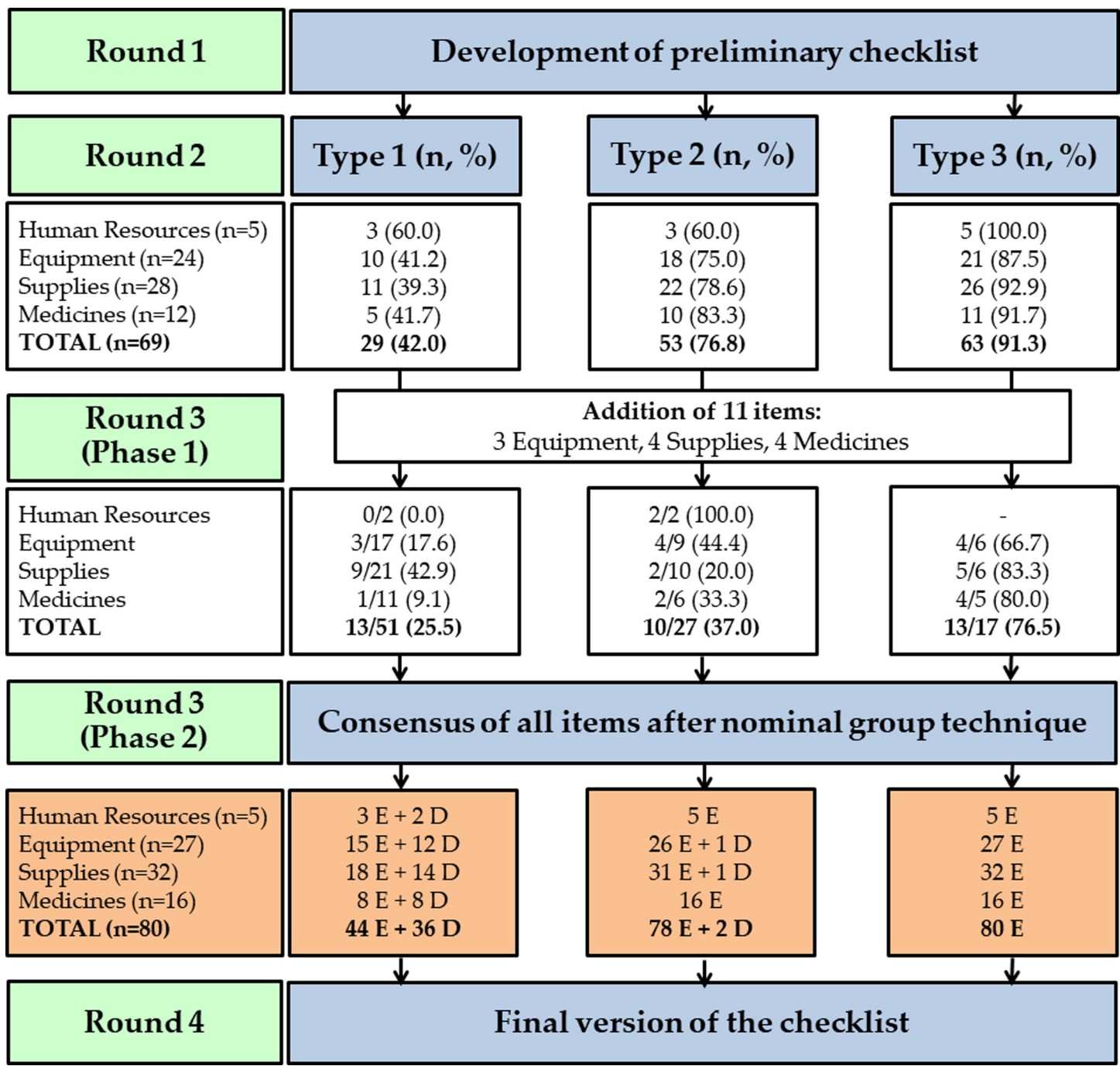

**Fig 3. Proportion of items validated by expert judges in each round.**

### Round 3: Two-phase nominal group technique

**Participant characteristics.**   A total of 12 potential experts were invited to participate in the nominal group meeting. Two invitations were refused. 10 potential experts accepted and met the expert criteria (Table 6). Half of the expert judges were men (50%). The average age was 46.1 years. Professions included physicians (40%), nurses (40%), and pharmacists (20%).

**Table 4. Section overview of the first version of the checklist.**

| Section | Description | Number of items |
|---|---|---|
| Health Unit Information | Unit identification, location/address, hours of operation, means of transportation and communication, electricity capacity, contact person information, and more | - |
| Human Resources[*] | Availability of the different professional categories and work schedules, numbers of personnel in each professional category, and more | 5 |
| Equipment | Equipment potentially used for SBE patient care and antivenom storage (e.g., pulse oximeter, stretcher) | 24 |
| Supplies | Medical and other supplies potentially used for SBE patient care and antivenom storage (e.g., sterile gloves, bandages) | 28 |
| Medicines | Medicines potentially used for SBE patient care (e.g., antibiotics, sedatives) | 12 |

[*] In Round 2, expert judges were asked to evaluate whether each professional category should have been trained in the clinical management of snakebites and/or storage of antivenom.

All expert judges were from Amazonas State. Professional experience averaged 19.4 years, with most expert judges holding advanced degrees (100%), previous or current professor positions in universities (80%), published articles (70%), and clinical SBE care experience (90%). Average expert criteria score was 12.3. See S4 File for additional characteristics.

**Nominal group meeting.   Phase 1.** A full list of unvalidated and added items from the survey round is provided in S4 File. Most of the unvalidated items reached consensus in Type 3 (76.5%) units. However, expert judges did not reach consensus on the majority of items in Type 1 (74.5%) and Type 2 (63.0%) centers. Overall, in Type 1, the sections with the lowest JPA items were Human Resources (0.0%), Medicines (9.1%), and Equipment (17.6%). In Type 2, items in Supplies (20.0%) had the lowest JPA. In Type 3, items in Equipment (66.7%) had the lowest JPA. Of the items validated in Phase 1, seven were considered Essential and six were considered Desirable for Type 1 community health centers. All items validated for Type 2 and

**Table 5. Characteristics of expert judges in the survey round.**

| Characteristics | Expert judges (N = 20) |
|---|---|
| **Age[1]** (yrs) | 44.8 (10.9) |
| **Gender[2]** | |
| Male | 12 (60%) |
| Female | 8 (40%) |
| **Profession[2]** | |
| Physician | 5 (25%) |
| Nurse | 15 (75%) |
| **Professional experience[1]** (yrs) | 11.9 (7.6) |
| **Practicing state[2]** | |
| Amazonas | 10 (50.0%) |
| Acre | 1 (5.0%) |
| Rondônia | 2 (10%) |
| Roraima | 2 (10%) |
| São Paulo | 3 (15%) |
| Federal District | 2 (10%) |
| **Expert criteria score[1]** | 10.90 (2.61) |

[1] Mean (SD)

[2] N (%)

**Table 6. Characteristics of expert judges in the nominal group meeting.**

| Characteristics | Expert judges (N = 10) |
|---|---|
| **Age**[1] (yrs) | 46.1 (10.2) |
| **Gender**[2] | 5 (50.0%) |
| Male | 5 (50.0%) |
| Female | |
| **Profession**[2] | 4 (40.0%) |
| Physician | 4 (40.0%) |
| Nurse | 2 (20.0%) |
| Pharmacist | |
| **Professional experience**[1] (yrs) | 19.4 (7.8) |
| **Practicing state**[2] | 10 (100.0%) |
| Amazonas | |
| **Expert criteria score**[1] | 12.3 (1.4) |

[1] Mean (SD)

[2] N (%)

Type 3 were considered Essential. All items suggested by survey round expert judges were added for a total of 80 items.

**Phase 2.** Anonymous Phase 1 results were fed back to the expert judges. The open discussion focused on Essential versus Desirable item classification in Type 1 community health centers, with some conversation on item priority in Type 2 centers as well. Judges raised concerns that without certain items, Type 1 centers would not be able to adequately treat potential adverse reactions from antivenom. Most judges, however, advocated that the function of Type 1 centers is early treatment, including antivenom, and patients can be transferred to higher level care (Type 2 or Type 3) in the rare case of an adverse reaction. After discussion, all expert judges agreed with this opinion, and classified 36 items (45.0%) as Desirable and 44 items (55.0%) as Essential in Type 1. However, in Type 2 centers, expert judges classified most items as Essential (97.5%). Expert judges considered all 80 items (100%) as Essential in Type 3 units. No items were considered Unessential or Undesirable. Only one nominal group meeting was required to reach JPA ≥75% on all items.

## Round 4: Checklist finalization and closing criteria

The final lists of essential and desirable items, respectively, for all three types of health units are outlined in Table 7.

With this final list, the expert steering committee outlined which procedures each type of health unit should be equipped to perform (Table 8). The Essential items considered for Type 1 centers are sufficient for antivenom premedication, storage, and administration, whereas a significant number of items required for treatment of early adverse reactions, management of complications, patient follow-up, and patient accommodation were considered Desirable. Type 2 centers had two or less items considered Desirable for each procedure. Type 3 units had all items for these procedures classified as Essential.

The final minimum requirements checklist is available in English and Portuguese in S5 File.

## Discussion

To our knowledge, this study is the first to determine the minimum supplies and staff required for safe and effective antivenom administration in community health centers, and, in doing so, develop a standardized checklist to evaluate health unit capacity for antivenom. Our results

**Table 7. Essential and desirable items in the checklist by each type of health unit.**

| Item | Essential (E) | | | Desirable (D) | | |
|------|------|------|------|------|------|------|
| | Type 1 | Type 2 | Type 3 | Type 1 | Type 2 | Type 3 |
| **Human Resources**[*] | | | | | | |
| Nursing technician/nursing assistant | X | X | X | | | |
| Registered nurse | X | X | X | | | |
| Physician | X | X | X | | | |
| Clinical analysis laboratory | | X | X | X | | |
| Pharmacist | | X | X | X | | |
| **Equipment** | | | | | | |
| Pulse oximeter | X | X | X | | | |
| Clinical thermometer | X | X | X | | | |
| Sphygmomanometer | X | X | X | | | |
| Bag-valve-mask | X | X | X | | | |
| Intubation kit for children | | X | X | X | | |
| Intubation kit for adults | | X | X | X | | |
| Stretcher | X | X | X | | | |
| Stretcher trolley | | X | X | X | | |
| Hospital armchair | | X | X | X | | |
| Intravenous infusion pole | X | X | X | | | |
| Phlebotomy armrest | X | X | X | | | |
| Defibrillator | | X | X | X | | |
| Vaccine refrigerator | X | X | X | | | |
| Domestic type refrigerator | X | X | X | | | |
| Refrigerator thermometer | X | X | X | | | |
| Oxygen cylinder | X | X | X | | | |
| Oxygen flowmeter | | X | X | X | | |
| Water bath | X | X | X | | | |
| Glass tubes | X | X | X | | | |
| Emergency trolley | | X | X | X | | |
| Stethoscope | X | X | X | | | |
| Ice pack | X | X | X | | | |
| Styrofoam box | X | X | X | | | |
| Wheelchair | | X | X | X | | |
| Heart monitor | | X | X | X | | |
| Reclining stretcher | | X | X | X | | |
| Hospital screen | | | X | X | X | |
| **Supplies** | | | | | | |
| Syringes 1 mL | X | X | X | | | |
| Syringes 3 mL | | X | X | X | | |
| Syringes 5–20 mL | X | X | X | | | |
| Flexible peripheral venous catheter for children | X | X | X | | | |
| Flexible peripheral venous catheter for adults | X | X | X | | | |
| Rigid peripheral venous catheter for children | X | X | X | | | |
| Rigid peripheral venous catheter for adults | X | X | X | | | |
| Cotton wool | X | X | X | | | |
| Gauze | X | X | X | | | |
| Multi-way or 3-way tap | | X | X | X | | |
| O2 catheter | X | X | X | | | |

(*Continued*)

**Table 7.** (*Continued*)

| Item | Essential (E) | | | Desirable (D) | | |
|---|---|---|---|---|---|---|
| | Type 1 | Type 2 | Type 3 | Type 1 | Type 2 | Type 3 |
| Needles (13x4.5) | | X | X | X | | |
| Needles (25x7) | X | X | X | | | |
| Needles (25x8) | X | X | X | | | |
| Medical tape | X | X | X | | | |
| Tourniquet for blood collection | X | X | X | | | |
| Non-sterile gloves | X | X | X | | | |
| Sterile gloves | | X | X | X | | |
| Measuring tape | | X | X | X | | |
| Skin marker | | X | X | X | | |
| Bandage | X | X | X | | | |
| Macrodrip IV infusion set | X | X | X | | | |
| Micro Drip IV infusion set | X | X | X | | | |
| Scalpel blade | | X | X | X | | |
| Oxygen mask | | X | X | X | | |
| Disposable surgical mask | | X | X | X | | |
| Suture kit[#] | | X | X | X | | |
| Penrose drain | | X | X | X | | |
| Central venous catheter | | X | X | X | | |
| Indwelling urinary catheter | | X | X | X | | |
| Urine collection bag | | X | X | X | | |
| Laryngeal mask airway | | | X | X | X | |
| **Medicines** | | | | | | |
| Corticosteroids | X | X | X | | | |
| Anti-histamines | X | X | X | | | |
| Adrenaline | X | X | X | | | |
| Painkillers | X | X | X | | | |
| Opioids | | X | X | X | | |
| Diuretic | | X | X | X | | |
| Saline 0.9% | X | X | X | | | |
| Glucose solution 5% | X | X | X | | | |
| Antibiotics | | X | X | X | | |
| Sedatives | | X | X | X | | |
| Topical anesthetics | | X | X | X | | |
| Antiseptics | X | X | X | | | |
| Anticonvulsivantes | | X | X | X | | |
| Bicarbonate | | X | X | X | | |
| Antiemetics | X | X | X | | | |
| Atropine | | X | X | X | | |

\* Professional categories in which snakebite envenoming training is Essential or Desirable

\# Suture kit includes scissors, tweezers, nylon thread 3.0 / cotton 0.2, and scalpel

highlighted three key findings: (1) the definition of Type 1 health centers and its list of Essential items by expert judges is consistent with the Brazilian Ministry of Health, WHO snakebite strategic plan, and a general snakebite capacity guideline in India (internal validity), (2) the list of Essential items for antivenom administration and clinical management is feasible and aligns

**Table 8. Summary item classifications as Essential or Desirable by clinical SBE procedure.**

| Clinical procedure | Number of items needed | Essential (E) | | | Desirable (D) | | |
|---|---|---|---|---|---|---|---|
| | | Type 1 | Type 2 | Type 3 | Type 1 | Type 2 | Type 3 |
| Antivenom premedication | 22 | 17 | 22 | 22 | 5 | 0 | 0 |
| Treatment of early adverse reactions | 31 | 13 | 29 | 31 | 14 | 2 | 0 |
| Antivenom administration | 18 | 16 | 18 | 18 | 2 | 0 | 0 |
| Management of complications | 33 | 13 | 32 | 33 | 20 | 1 | 0 |
| Patient follow-up | 12 | 8 | 12 | 12 | 4 | 0 | 0 |
| Antivenom storage | 5 | 5 | 5 | 5 | 0 | 0 | 0 |
| Patient accommodation | 5 | 1 | 4 | 5 | 4 | 1 | 0 |

with the literature regarding snakebite clinical care (reliability), and (3) engagement of local experts is critical to developing and implementing an antivenom decentralization strategy (feasibility).

## Internal validity: Snakebite envenoming care in community health centers

Our results defined community health centers as primary care clinics capable of providing emergency care to snakebite envenoming patients, including antivenom, with the capacity to refer all or severe patients to a higher-level unit after antivenom administration. Health unit capacity was further defined by the minimum Human Resources, Equipment, Supplies, and Medicines outlined in the final checklist for Type 1.

These results align with the Brazilian MoH definition of CHCs and their scope of practice. The MoH states CHCs are primary health facilities with a small team, usually one doctor, nurse, nursing assistant (matches Essential items in Type 1 centers), and at least four community health workers (CHW) [41]. CHWs were not included in the checklist—likely due to its explicit focus on clinical care. In terms of practice, the MoH states services provided by CHCs include preventative care, public health interventions, maternal and child health care, management of chronic non-communicable diseases, referrals to higher-level care as well as social, sanitation, and other services [42,43]. The Equipment, Supplies, and Medicines items listed in the checklist, excluding antivenom, fall under those required to perform these services [44]. Further, in 2020, the SUS launched the Requalifica Programme to construct, expand, and refurbish CHCs with the goal of ensuring adequate infrastructure [45]. This aligns with the Basic Health Information section of the checklist requiring a regular power supply.

Our results are also in accordance with the WHO definition of CHCs and their capacity. The WHO strategy for prevention and control of SBE specifically aims to improve "access to essential medicines, including antivenoms, and all other medical drugs, equipment, and consumable items" and ensure "appropriate staffing" in primary health care services [46]. Our checklist categories—Human Resources, Medicines, Equipment, and Supplies—match these target areas. The WHO strategy, however, does not specify the essential items recommended for snakebite care in CHCs. This study adds to the literature in offering a detailed perspective on the minimum requirements for safe, effective antivenom administration and clinical management of snakebites in a low-resource, high-burden region.

One other study in India, although general, also provides perspective on this tenet of the WHO strategy. A team of researchers utilized data from a population-linked facility survey conducted by the MoH and Family Welfare, Government of India, to conceptualize structural capacity for snakebite care [18]. This assessment is largely like our checklist, but differed in that it was more conceptual and had a focus beyond clinical care in its inclusion of two

additional categories: Governance and Finance and Health Management Information Systems. After explaining its assessment model, the study explicitly called for a specific health facility survey (like the checklist developed) to assess SBE care capacity [18].

## Reliability: Essential items for antivenom administration and clinical management

The debate regarding essential items for Type 1 centers parallels the data and health professional opinions present in the literature, specifically the critical need for antivenom in CHCs, longstanding fear of adverse reactions to antivenom, and additional training for health professionals. This concordance suggests the data informing the checklist is reliable.

There was no debate among expert judges regarding whether antivenom was essential in all three types of health units. Several other studies among professionals in the Brazilian Amazon ubiquitously argue antivenom as an essential medicine in CHCs [2,4,5,8,11,13–15,17,19,47,48]. In addition, research on snakebite envenoming care across India [49,50], French Guiana [51], Burkina Faso [52], and Myanmar [53] highlights the need for antivenom availability and accompanying equipment in their primary health care centers. A multi-country study regarding access to antivenom in Malaysia, Thailand, Indonesia, Philippines, Vietnam, Lao PDR, and Myanmar also explicitly argues "strengthening the supply chain of antivenoms to ensure that antivenoms are readily accessible at the point of service," primary health units [54].

While antivenom is internationally recognized as an essential medicine in primary care, some expert judges in our study raised serious concerns regarding the capacity of CHCs to adequately treat potential early adverse reactions to antivenom. This fear has been documented in previous Brazilian Amazon [19] and rural India [55] studies and likely stems from the more frequent and severe reactions observed with older, pre-2000s antivenoms that required hospital treatment [56,57]. With newer, improved antivenoms, the frequency of early adverse reactions is low - around 15% in the Brazilian Amazon [58] - and almost always a mild skin reaction as reported by studies in Brazil [58] and Costa Rica [59]. It is worth noting that whether items required for treating reactions were essential to Type 1 CHCs was the largest obstacle to reaching consensus amongst the expert judges. The ultimate decision to not include these items was only accepted given the existing referral pathway from CHCs (Type 1, Type 2) to hospitals (Type 3).

The reluctance of some judges to accept leveraging lower-level units for time-sensitive treatment, even with referral networks in place, is likely due to the hospital-centric narrative and training surrounding snakebite envenoming care. This is supported by how easily the judges reached consensus on essential items in Type 3 health units, or hospitals. But, when asked to determine Essential versus Desirable items for the lower-level Type 2 and Type 1 centers, there was significantly more disagreement and debate surrounding whether patient care can be extended from hospitals to earlier along the care continuum despite its safety and proven effectiveness in other settings, namely antivenom decentralization in Costa Rica [59] and a pilot trial of a nurse-led antivenom clinic Tanzania [60].

This is reflected in the expert judges' call for specifically trained physicians, nurses, and nursing assistants in CHCs, also in accordance with the literature in Southeast Asia [54], Nigeria [61], India [55], and Burkina Faso [52]. In contrast with this view, several studies also highlight lack of knowledge and call for additional, reformed training amongst hospital-based health providers, in particular the United States [62] and Bhutan [63]. Most decentralization initiatives recommend training for the primary health care providers, but, considering the lengthy debate between expert judges in this study and emerging literature on potential gaps

in SBE training for hospital-based providers, we suggest decentralizing training programs across the continuum of care. Hospital-centric curricula, nor a sole CHC focus, are adequate to address a severe, time-sensitive disease predominantly occurring in remote and rural areas.

## Feasibility: Engagement of local experts for successful decentralization planning

A modified-Delphi method engaging local experts was necessary to develop a valid and reliable minimum requirements checklist that is feasible within the current health unit structure in the Brazilian Amazon. Without engaging local experts, 11 items (notably only one of which considered essential in Type 1 centers) would not have been added to the checklist, nor would training across the care continuum—specifically how to address early adverse reactions—been identified as a priority to antivenom decentralization strategies.

Several studies have argued the success of health care decentralization programs is highly dependent on the context [64]. Influencing contextual factors include cultural norms, values, practices, and beliefs; geographic environment; and a well-functioning health system, including logistics support, supplies, and equipment [65]. A study on decentralization and health system performance in India identified three determinants of performance: health workers, health facilities, and agents of decision making, patients, and the community [66].

Regarding health workers and agents of decision-making, care coordination, training, and support for CHC health professionals is critical to decentralization success. Our team has been conducting trainings with health professionals as funding allows, and established both an online group and an SMS-based care coordination messaging system to connect professionals with each other as well as experts at our institute/hospital, the Foundation for Tropical Medicine in Manaus. The online group includes doctors, pharmacists, nurses, and even biologists to support these health professionals real-time in the identification and management of snakebites, from determining the snake species to medication administration. The SMS-based messaging system connects professionals with an expert at our institute/hospital to answer any questions, also in real-time. In addition, health professionals in Type 1 and Type 2 units often communicate with each other via radio or telephone, and engage senior staff to support and advise in snakebite envenoming cases.

Narrowing in on health facilities, the study detailed the "availability of infrastructure, equipment, and supplies" as well as "accreditation status" as critical components [66]. The checklist developed addresses these two components and represents an initial milestone for determining the absolute minimum supplies and staff necessary for safe and effective antivenom administration and clinical management. Given the structure of the Brazilian health system is already decentralized to municipalities, and the essential items align with the current capacity of CHCs, antivenom decentralization is feasible.

## Limitations

As with all Delphi studies, the quality of data collected is tied to the qualifications and experience of the participating experts. To obtain the most robust data possible, we engaged three groups of expert judges across four rounds of iteration, and expert qualifications, experience, and demographics were disclosed in detail to promote transparency. A potential bias exists in the lack of health professionals actively working in community health centers. The objective of the study, however, was to determine the essential items to safe and effective clinical care of snakebites. Clinical and clinical research experts were thus engaged.

## Conclusion

Decentralization of antivenom access is a set of strategic actions to reduce mortality and morbidity from snakebite envenoming, primarily affecting historically neglected and invisible populations. This study joins an international set of evidence advocating for decentralization, adding value in its definition of essential care items; identification of training needs across the care continuum; and demonstration of the validity, reliability, and feasibility provided by engaging local experts. Specific to Brazil, further added value comes in the potential use of the checklist for health unit accreditation as well as its applications to logistics and resource distribution. Future research priorities should apply this checklist to CHCs in the Amazon region of Brazil to determine which CHCs are or could be capable of receiving antivenom and translate this expert- driven checklist and approach to snakebite care in other settings or other diseases in low-resource settings.

## Supporting information

**S1 File. Version of the minimal requirements checklist created by the steering committee for experts' validation.**
(DOCX)

**S2 File. Characterization of the 20 expert judges responsible by the first phase of the validation process of the minimal requirements checklist.**
(XLSX)

**S3 File. Joint-probability of agreement for each item by type of unit.**
(XLSX)

**S4 File. Characterization of the 10 expert judges responsible by the second phase of the validation process (by nominal group technique) of the minimal requirements checklist.**
(XLSX)

**S5 File. Final version of the minimal requirements checklist.**
(DOCX)

## Acknowledgments

We would like to extend a special thanks to the experts who kindly agreed to participate in this study. We know the time of these professionals is very valuable, and that makes us especially grateful.

## Author Contributions

**Conceptualization:** Anna Tupetz, Flávia Ramos, Elizabeth Teixeira, Catherine Staton, João Vissoci, Charles J. Gerardo, Fan Hui Wen, Jacqueline Sachett, Wuelton M. Monteiro.

**Data curation:** Eleanor Strand, Joseir Saturnino, Wuelton M. Monteiro.

**Formal analysis:** Eleanor Strand, Joseir Saturnino, Aline Alencar, Wuelton M. Monteiro.

**Funding acquisition:** João Vissoci, Charles J. Gerardo, Jacqueline Sachett, Wuelton M. Monteiro.

**Investigation:** Gisele Rocha, Altair Seabra de Farias, Aline Alencar, Anna Tupetz, Flávia Ramos, Elizabeth Teixeira, João Vissoci, Fan Hui Wen, Jacqueline Sachett, Wuelton M. Monteiro.

**Methodology:** Gisele Rocha, Joseir Saturnino, Altair Seabra de Farias, José Diego Brito-Sousa, Flávia Ramos, Elizabeth Teixeira, Charles J. Gerardo, Fan Hui Wen, Jacqueline Sachett, Wuelton M. Monteiro.

**Project administration:** Thiago Serrão-Pinto, André Sachett, Catherine Staton, Wuelton M. Monteiro.

**Resources:** André Sachett, Wuelton M. Monteiro.

**Software:** Thiago Serrão-Pinto, André Sachett, Wuelton M. Monteiro.

**Supervision:** Jacqueline Sachett, Wuelton M. Monteiro.

**Validation:** Thiago Serrão-Pinto, Eleanor Strand, Wuelton M. Monteiro.

**Visualization:** Thiago Serrão-Pinto, Eleanor Strand, Wuelton M. Monteiro.

**Writing – original draft:** Thiago Serrão-Pinto, Eleanor Strand, Wuelton M. Monteiro.

**Writing – review & editing:** Gisele Rocha, André Sachett, Joseir Saturnino, Altair Seabra de Farias, Aline Alencar, José Diego Brito-Sousa, Anna Tupetz, Flávia Ramos, Elizabeth Teixeira, Catherine Staton, João Vissoci, Charles J. Gerardo, Fan Hui Wen, Jacqueline Sachett.

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
