## [Decision Letter · Decision Letter 0]

20 Nov 2023

Dear Dr Monteiro,

Thank you very much for submitting your manuscript "Development and validation of a minimum requirements checklist for snakebite envenoming treatment in the Brazilian Amazonia" for consideration at PLOS Neglected Tropical Diseases. As with all papers reviewed by the journal, your manuscript was reviewed by members of the editorial board and by several independent reviewers. In light of the reviews (below this email), we would like to invite the resubmission of a significantly-revised version that takes into account the reviewers' comments. 

Considering the comments received form the 3 reviewers, we would like to recommend a significant level of revision by addressing all the reviewers comments carefully.

We cannot make any decision about publication until we have seen the revised manuscript and your response to the reviewers' comments. Your revised manuscript is also likely to be sent to reviewers for further evaluation.

Sincerely,

Kalana Prasad Maduwage, MBBS, MPhil, PhD, FRSPH (UK), FRCP (Edin)

Academic Editor

José María Gutiérrez

Section Editor

Considering the comments received form the 3 reviewers, we would like to recommend a significant level of revision by addressing all the reviewers comments carefully.

Reviewer's Responses to Questions

**Key Review Criteria Required for Acceptance?**

**Methods**

-Are the objectives of the study clearly articulated with a clear testable hypothesis stated?

-Is the study design appropriate to address the stated objectives?

-Is the population clearly described and appropriate for the hypothesis being tested?

-Is the sample size sufficient to ensure adequate power to address the hypothesis being tested?

-Were correct statistical analysis used to support conclusions?

-Are there concerns about ethical or regulatory requirements being met?

Reviewer #1: The methods are adequate!

Reviewer #2: The methods used are mostly clearly described.

However, I don't understand the differentiation of Community Health Centers (CHCs) into Type 1,2 and 3 in Table 2

Type 1 seems to be a CHC for basic primary health care, but Type 2 and 3 seem to meet the requirements of a hospital. In the discussion section it is written once that Type 3 CHC are in fact hospitals. If a CHC is capable to treat all complications of snakebite envenoming in the Amazon region (Type 3 CHC), which would include respirator treatment, dialysis for Acute Kidney Injury (AKI), blood transfusion, surgical interventions for local cytotoxic venom effects etc. then it is already a referral and even tertiary hospital. This needs clarification.

Reviewer #3: -Are the objectives of the study clearly articulated with a clear testable hypothesis stated? Yes

-Is the study design appropriate to address the stated objectives? Yes

-Is the population clearly described and appropriate for the hypothesis being tested? N/A

-Is the sample size sufficient to ensure adequate power to address the hypothesis being tested? N/A

-Were correct statistical analysis used to support conclusions? N/A

-Are there concerns about ethical or regulatory requirements being met? No

**Results**

-Does the analysis presented match the analysis plan?

-Are the results clearly and completely presented?

-Are the figures (Tables, Images) of sufficient quality for clarity?

Reviewer #1: The results were achieved in accordance with the proposed objective!

Reviewer #2: For the checklist of required equipment I would add a device (Monitor) for continuous ECG and heart rate monitoring, blood pressure and Oxygen saturation. This is at least desirable for Type 1 CHCs and essential for Type 2 and 3 CHC (in fact hospitals). This facilitates monitoring of the patients during antivenom administration, which is most important.

In the Amazon region Bothrops and Lachesis species are the most important snake species causing > 90% of envenoming. They cause cytotoxic and haematotoxic symptoms and for the indication of antivenom administration I need a minimum of lab results (Blood cell count and 20 minute whole blood clotting test). I don't see any equipment for the possibility to perform these tests, neither in Type 1 nor in type 2 and 3 CHC.

Reviewer #3: -Does the analysis presented match the analysis plan? Yes

-Are the results clearly and completely presented? yes

-Are the figures (Tables, Images) of sufficient quality for clarity? yes

**Conclusions**

-Are the conclusions supported by the data presented?

-Are the limitations of analysis clearly described?

-Do the authors discuss how these data can be helpful to advance our understanding of the topic under study?

-Is public health relevance addressed?

Reviewer #1: The discussion was well developed according to the research results.

The conclusions are relevant and contribute to the need for training health professionals in relation to the topic of snakebites.

Reviewer #2: Decentralized supply of Antivenom is an important step forward to improve management of snakebite envenoming particularly in regions where transport to the next hospital is difficult and time consuming. The authors suggest a checklist to ensure safe administration of antivenom. However in the discussion section the importance of well trained health care personnel falls a bit short. Beside equipment, knowledge and experience of health care staff is crucial to manage snakebite victims and avoid death due to adverse reactions during antivenom administration. This cannot be achieved by short training courses, but need practical experience. A consultation service provided by experienced clinicians would help to support staff in CHCs. Snake identification, dealing with initial life-threatening events (hypovolaemic shock, respiratory failure), indication for antivenom, treatment of side effects need to be addressed. Health care staff in CHC have usually no experience at all in management of snakebite envenoming and need to have a contact to ask questions and discuss a case in real time. The importance of a consultation service should be mentioned in the discussion part. It will help to improve knowledge and experience of those in charge at CHCs. Does a Poison Control Center exists in Manaus or elsewhere in Brazil? This would be the Institution to be contacted and give advice. This is the case for example in Thailand where the Poison Control Center in Bangkok can be contacted and asked for advice. At the same time data on snakebites in the region can be collected.

The administration of antivenom has a life-threatening side effect (anaphylactic shock) and there is concern that snakebite victims may die from antivenom side effects. We would do more harm than good.The possibility for health care staff in CHC's to contact experts is therefore absolutely mandatory before a decision to give antivenom is made and to deal with side effects. The staff in CHC should not be left alone with these decisions! 

In the discussion section there is one sentence that I don't understand: "It is worth noting that whether items required

for treating reactions were essential to Type 1 CHCs was the largest obstacle to reaching consensus amongst

the expert judges. The ultimate decision to not include these items was only accepted given the existing

referral pathway from CHCs to hospitals." Actually the CHCs Type 1 are the health facilities where AV is supposed to be used and then all items should be included to treat adverse reactions. Adrenalin the essential drug to treat anaphylaxis is included, and antihistamines too. Which items where not included?

Reviewer #3: -Are the conclusions supported by the data presented? Yes

-Are the limitations of analysis clearly described? Yes

-Do the authors discuss how these data can be helpful to advance our understanding of the topic under study? Yes

-Is public health relevance addressed? Yes

**Editorial and Data Presentation Modifications?**

Reviewer #1: The manuscript with few corrections is attached!

Evaluate the use of the terms poison, venom, poisonous and venomous.

Reviewer #2: no modification necessary

Reviewer #3: None

**Summary and General Comments**

Reviewer #1: This is well-conducted research and is extremely important for Sudan, as well as being a reference for other similar studies to be developed in other regions and countries where snakebites represent an important cause of morbidity and mortality.

Reviewer #2: The Remote Envenoming Consultation Service which has been organized in Malaysia is a very good example, where experts guide less experienced health care staff to manage snakebite patients and ensure state of the art treatment of snakebite envenoming. A group of consultants located in different regions support health care staff in district hospitals and Primary Health Centers in snakebite management 24/7 in real time by using smart phones. A contact number should be available in every CHC in the Amazon region that is suppose to treat snakebite envenoming including the administration of AV. A Poison Control Center or Tertiary Hospital would be suitable to give advice.

Comment on Supporting information: the authors list 6 Supporting Information files but I have seen only five.

Reviewer #3: The manuscript titled “Development and validation of a minimum requirements checklist for snakebite envenoming treatment in the Brazilian Amazonia.’ Describes a study that developed and validated a checklist to evaluate the minimum requirements for community health centers in the Amazon region of Brazil to adequately treat snakebite envenoming, using modified-Delphi technique. This is a very important, pragmatic study, and the manuscript is well-written. I have minor comments for the improvement of the manuscript.

The Introduction should provide some orientation for the international readers on the epidemiology of snakebite, available antivenoms and severe acute adverse reaction to antivenom in the Brazilian Amazon. In particular, briefly describing (at least by a diagram) the network of healthcare facilities of different levels that involve in snakebite management, their staff involved in managing patients (e.g. whether all these centers are manned by qualified medical doctors and, nurses with emergency care training) would be important in understanding the context. 

“In the abstract and introduction, the first sentence; “Currently, antivenoms are the only evidence-based, safe, and effective treatment for snakebite envenoming. This is quite an overstatement. As recent systematic reviews revealed there are no randomised placebo-controlled trials that have shown any effectiveness of antivenoms for the main systemic effects of envenoming such as coagulopathy, neurotoxicity or acute kidney injury. This means there is no highest-quality evidence for the effectiveness of antivenom. However, this doesn't mean that antivenoms are not working at all. Studies that showed the effectiveness of antivenom without a patient group that did not receive antivenom are very few. In addition,, antivenoms used in some regions of the world are associated with very high adverse reaction rates (E.g. Indian polyvalent antvenom). A statement like “Currently, antivenoms are the only specific treatment available for snakebite envenoming” would be more realistic.

PLOS authors have the option to publish the peer review history of their article (what does this mean?). If published, this will include your full peer review and any attached files.

Reviewer #1: Yes: Paulo Sérgio Bernarde

Reviewer #2: Yes: Jörg Blessmann

Reviewer #3: No

Figure Files:

Data Requirements:

Please note that, as a condition of publication, PLOS' data policy requires that you make available all data used to draw the conclusions outlined in your manuscript. Data must be deposited in an appropriate repository, included within the body of the manuscript, or uploaded as supporting information. This includes all numerical values that were used to generate graphs, histograms etc.. For an example see here: http://www.plosbiology.org/article/info:doi%2F10.1371%2Fjournal.pbio.1001908#s5.
---

## [Decision Letter · Decision Letter 1]

15 Jan 2024

Dear Dr Monteiro,

We are pleased to inform you that your manuscript 'Development and validation of a minimum requirements checklist for snakebite envenoming treatment in the Brazilian Amazonia' has been provisionally accepted for publication in PLOS Neglected Tropical Diseases.

Best regards,

Kalana Prasad Maduwage, MBBS, MPhil, PhD, FRSPH (UK), FRCP (Edin)

Academic Editor

José María Gutiérrez

Section Editor

Both reviewers and the editorial have accepted the revised version of the paper based on the successfully addressing and editing the manuscript.

Reviewer's Responses to Questions

**Key Review Criteria Required for Acceptance?**

**Methods**

-Are the objectives of the study clearly articulated with a clear testable hypothesis stated?

-Is the study design appropriate to address the stated objectives?

-Is the population clearly described and appropriate for the hypothesis being tested?

-Is the sample size sufficient to ensure adequate power to address the hypothesis being tested?

-Were correct statistical analysis used to support conclusions?

-Are there concerns about ethical or regulatory requirements being met?

Reviewer #2: The objectives of the study are clearly articulated. The types of Health Units are clearly and better defined.

Reviewer #3: No further comments

**Results**

-Does the analysis presented match the analysis plan?

-Are the results clearly and completely presented?

-Are the figures (Tables, Images) of sufficient quality for clarity?

Reviewer #2: Results are clearly presented and match the analysis plan. Figure 1 was added and visualize clearly the management flow for better understanding.

Reviewer #3: No further comments

**Conclusions**

-Are the conclusions supported by the data presented?

-Are the limitations of analysis clearly described?

-Do the authors discuss how these data can be helpful to advance our understanding of the topic under study?

-Is public health relevance addressed?

Reviewer #2: Yes, beside the essential and necessary equipment list for the safe administration of antivenom and the potential use for health unit accreditation, the importance of real time consultation for the decentralization success has been added in the discussion section.

Reviewer #3: No further comments

**Editorial and Data Presentation Modifications?**

Reviewer #2: No modification

Reviewer #3: No further comments

**Summary and General Comments**

Reviewer #2: The revision significantly improved the manuscript

Reviewer #3: Authors have addressed my comments. I have no further comments.

PLOS authors have the option to publish the peer review history of their article (what does this mean?). If published, this will include your full peer review and any attached files.

Reviewer #2: **Yes: **Jörg Blessmann

Reviewer #3: **Yes: **Anjana Silva

---

## [Editor Report · Acceptance letter]

17 Jan 2024

Dear Dr. Monteiro,

We are delighted to inform you that your manuscript, "Development and validation of a minimum requirements checklist for snakebite envenoming treatment in the Brazilian Amazonia," has been formally accepted for publication in PLOS Neglected Tropical Diseases.

Best regards,

Shaden Kamhawi

co-Editor-in-Chief

Paul Brindley

co-Editor-in-Chief
